# Assessing the Impact of Transcutaneous Maxillary Distraction Osteogenesis on Pharyngeal Airway Volume in Children with Cleft Lip and Palate: A 3D Evaluation Study

**DOI:** 10.3390/jpm13030543

**Published:** 2023-03-18

**Authors:** Chia-Hsuan Chan, Chi-Yu Tsai, Jui-Pin Lai, Shiu-Shiung Lin, Yu-Jen Chang

**Affiliations:** 1Department of Craniofacial Orthodontics, Department of Dentistry, Kaohsiung Chang Gung Memorial Hospital and Chang Gung University College of Medicine, Kaohsiung 833, Taiwan; 2Department of Plastic and Reconstructive Surgery, Kaohsiung Chang Gung Memorial Hospital and Chang Gung University College of Medicine, Kaohsiung 833, Taiwan

**Keywords:** transcutaneous maxillary distraction osteogenesis (TMDO), airway volume, cleft lip, cleft palate, 3D

## Abstract

Cleft lip and cleft palate (CLCP) patients often have a retrusive maxilla and a severe skeletal Class III malocclusion, which can result in velopharyngeal insufficiency (VPI). The aim of this study was to evaluate the changes in the volume of the 3D airway in CLCP children after maxilla distraction using the transcutaneous maxillary distraction osteogenesis (TMDO) method. 15 children with bilateral or unilateral CLCP were included in the study. 3D CBCT images were taken before and after distraction and were segmented and reconstructed to create a 3D airway model. The airway was divided into three regions: the upper, oropharyngeal, and hypopharyngeal airway. Pearson correlation tests were used to assess correlations between volume changes and corresponding skeletal and dental landmark movements (Point N, ANS, A, B, Pog, U1, and L1). The results showed that the ANS point advanced 9.85 ± 3.60 mm, and the A point advanced 14.22 ± 4.57 mm. The total airway volume change increased by 2535.06 ± 2791.80 mm^3^. However, there was no significant correlation between the A/ANS/U1 and the three different airway regions. Only B/Pog/L1 showed a positive correlation with these airway regions, with a high correlation between B/Pog/L1 and the hypopharyngeal airway region. TMDO can result in greater anterior advancement of the maxilla and an increase in airway volume, but the changes in bony landmarks did not show a strong positive correlation with the increase in airway volume as expected. Further investigation is needed to analyze the influence of surrounding soft tissue on the changes in airway volume.

## 1. Introduction

Cleft lip and cleft palate (CLCP) is a common birth defect affecting the lip and maxilla region. Its causes may be influenced by ethnic, racial, geographic, and socioeconomic factors. The palate grows and fuses in two stages, the primary palate developing in the first six weeks and the secondary palate forming in the next 8 weeks [1,2,3,4]. The failure of fusion can lead to cleft formation and impaired mandibular development. The incidence of CLCP in Asia is 1.2–1.92 per 1000 births [5,6,7], with a male-to-female ratio of 1.19:1 according to a 2002 retrospective analysis of 991 Taiwanese children. Unilateral cleft lip is two times more common on the left side, and unilateral clefts are nine times more common than bilateral clefts [6,7,8,9]. According to the Clinical Study by Stuppia (2011) [10], syndromic CLCP is represented by Van der Woude syndrome (VWS, MIM 119300), which accounts for approximately 2% of all CLCP cases and is characterized by congenital sinuses of the lower lip. Non-syndromic CLCP is a multifactorial disease that arises from the interaction between genes and the environment, such as maternal exposure to smoke, alcohol, diet, viral infection, drugs, and teratogenic agents during early pregnancy.

As CLCP patients grow, maxillary retrusion is often observed due to reduced anterior-posterior maxillary growth resulting from scar tissue formation after surgical cleft lip repair. This can cause severe class III malocclusion and velopharyngeal insufficiency [11,12,13]. The upper airway volume in CLCP patients is significantly smaller compared to non-cleft patients and they may also have excessive nasal resonance during speech production [14,15,16]. A study by Chun-Shin Chang et al. in 2017 [15] suggested that cleft orthognathic surgery did not improve airway health or speech, and overnight polysomnographic studies should not be used as a presurgical assessment.

Distraction osteogenesis with a rigid external distraction device was first applied in 1997 by Polley and Figueroa [17] and is now widely used to treat CLCP patients with severe maxillary hypoplasia [18,19]. It provides an effective alternative to orthognathic surgery as it stimulates growth of both bony defects and soft tissue (Figure 1). Conventional orthognathic surgery to advance the maxilla is often more difficult and prone to relapse in CLCP patients due to severe scarring [20].

This study aims to evaluate the 3D volumetric changes in airway volumes of CLCP children after transcutaneous maxillary distraction osteogenesis and determine the correlation between skeletal landmarks and airway changes.

## 2. Material and Methods

Participants:

Fifteen patients with cleft lip and palate (CLCP) who underwent transcutaneous maxillary distraction osteogenesis (TMDO) treatment were included in this study. The participants consisted of 8 girls and 7 boys, with an average age of 11.43 ± 2 years old. Of the 15 patients, 5 had bilateral cleft lip and palate, while 10 had unilateral cleft lip and palate. All patients underwent primary lip and palate repair and alveolar bone graft at Kaohsiung Chung Lung Memorial Hospital (Table 1 and Table 2). These patients were diagnosed with Skeletal Class III pattern and were suggested to receive LeFort I osteotomy and TMDO treatment for improvement of mid-face hypoplasia.

Inclusive and Exclusive Criteria:

Inclusive criteria included: patient’s age between 10~13 y/o; unilateral or bilateral cleft/lip palate patient (non-syndromic type); and a patient with midface retrusion and malocclusion that require transcutaneous maxillary distraction osteogenesis (such patients provide consent for inclusion). Exclusive criteria included: presence of severe congenital facial asymmetry or traumatic deformity, and soft tissue airway surgery within 1 year.

All of our fifteen cases were non-syndromic type CLCP, as no cases showed evidence of Van der Woude syndrome [10].

Treatment:

The TMDO treatment was started 7 days after surgery, following the latency period. No additional surgical procedures, such as genioplasty, rhinoplasty, or infra-orbital augmentation, were performed. The distraction period lasted 1 month at a rate of 1 mm per day, and the devices were removed after 8–9 weeks of consolidation, once the desired mid-face improvement was achieved. All patients received continuous full-mouth orthodontic treatment.

Imaging:

Before starting the TMDO treatment, we took lateral cephalometric films and CBCT images of these 15 patients. The images were taken 1.5 months before the start of the distraction (T0) and 6–8 months after removal of the distraction devices (T1). We utilized medical CBCT for airway evaluation with patients in the lying position. This position was chosen as the airway dimension, and soft tissue are more similar to that in the sleeping condition. To standardize the patient’s head position, the 3D CBCT images were oriented with the Frankfort horizontal plane, which was defined as the plane passing through the bilateral orbitale and porion, and patients were all requested to assume a lying position with their tongues resting in relaxed posture. The sagittal plane was perpendicular to the Frankfort horizontal plane and the coronal plane. After taking the images, all CBCT scans were adjusted to be parallel to the Frankfort horizontal plane in the sagittal view.

Airway Analysis:

The airway changes were evaluated using a 3D airway modeling technique. The reference planes were parallel to the Frankfort horizontal plane, and the lowermost protruding parts of cervical vertebrae 1, 2, and 3 were defined as C1, C2, and C3, respectively. To evaluate different regions of the airway, we divided it into three regions, as shown in Figure 2:The upper airway: from the posterior nasal spine to the lowermost protruding part of C1.The oropharyngeal airway: from the lowermost protruding part of C1 to the lowermost protruding part of C2.The hypopharyngeal airway: from the lowermost protruding part of C2 to the lowermost protruding part of C3.

The researchers aimed to evaluate changes in the airway using a 3D modeling method. To do this, we utilized two open-source software programs, ITK-SNAP 3.6.0 and Slicer CMF. The segmentation tool in ITK-SNAP was used to create 3D models, and Slicer CMF was used to perform the necessary calculations. The models were created using the Model Maker module of Slicer CMF (Figure 3 and Figure 4) and were labeled using the labeling method. The airway was divided into three regions: the upper airway, the oropharyngeal airway, and the hypopharyngeal airway.

To assess changes in the airway volume, the researchers calculated the 3D distance changes over seven points, including Nasion, Anterior Nasal Spine, Point-A, Point-B, Skeletal Pogonion, Upper Central Incisor Edge, and Lower Central Incisor Edge (Table 3). In addition, they analyzed the FMPA angle changes in cephalometric films. The researchers then used Pearson correlation tests to evaluate the correlations between the volume changes in the upper, oropharyngeal, and hypopharyngeal airways and the distance changes in the surrounding facial skeletal landmarks.

All measurements were taken by one examiner to ensure consistency and reduce measurement errors. Overall, this method allows for a comprehensive evaluation of airway changes and provides valuable information for assessing and treating airway-related conditions.

## 3. Error Study

The 3D models’ landmarks were re-evaluated by a single researcher with a 1-month interval. The results showed no significant difference in defining skeletal landmarks during the tracing. We used Houston’s method (1983) [21], and the null hypothesis was tested, which indicated that there was no difference between the two measurements.

## 4. Results

We can see in Case-7,8 that the total airway volume showed a decrease, and in Case-9, although total airway volume showed an increase, it showed a decrease in the Oropharyngeal and the Hypopharngeal airway.

The average period of distraction was 30.53 ± 11.78 days, with a maintenance period of 68.67 ± 15.17 days (Table 2). The total average change in airway volume was increased by 2535.06 ± 2791.80 mm^3^ (Table 4 and Table 5), with increases in the upper, oropharyngeal, and hypopharyngeal airways of 946.33 ± 1628.85 mm^3^, 1091.32 ± 1032.81 mm^3^, and 497.41 ± 675.24 mm^3^, respectively. However, in two cases (Cases 7 and 8), there was a decrease in total and specific airway volumes. In Case 9, there was an increase in total airway volume but decreases in oropharyngeal and hypopharyngeal volumes.

The average changes in 3D skeletal landmarks were ANS (9.85 ± 3.60 mm), A (14.22 ± 4.57 mm), B (3.33 ± 1.60 mm), and Pog (3.54 ± 1.84 mm) (Table 6), and average FMPA angle changes in cephalometric film showed −1.08 ± 1.83 degrees. The average changes in 3D dental landmarks were U1 (16.77 ± 4.60 mm) and L1 (4.83 ± 1.37 mm).

The correlations between pharyngeal airway volume and skeletal/dental landmarks are shown in Table 7. In the upper airway groups, there were negligible correlations (0.00 < r < 0.30) between skeletal landmarks (N, ANS, A, B) and dental landmarks (U1, L1). There was low positive correlation (0.30 < r < 0.50) with Pog (r = 0.43). In the oropharyngeal airway groups, there were negligible correlations (0.00 < r < 0.30) between skeletal landmarks (N, ANS, A) and dental landmark U1. However, there were low positive correlations (0.30 < r < 0.50) with B (r = 0.37), Pog (r = 0.75), and L1 (r = 0.45). In the hypopharyngeal airway groups, there were negligible correlations (0.00 < r < 0.30) between N, ANS, A, and U1. There was moderate positive correlation (0.50 < r < 0.70) with L1 (r = 0.53) and high correlation (0.70 < r < 0.90) with B (r = 0.76) and Pog (r = 0.75).

When comparing the total airway volume change with landmarks, there were negligible correlations between N, ANS, A, and U1 (0.00 < r < 0.30), moderate positive correlation (0.50 < r < 0.70) with L1 (r = 0.53), and high correlations (0.70 < r < 0.90) with B (r = 0.77) and Pog (r = 0.75).

## 5. Discussion

Patients with cleft lip and palate (CLCP) have distinct craniofacial and airway structures compared to normal patients. Many CLCP patients with severe mid-face deficiencies receive distraction osteogenesis to improve their condition. However, no studies have compared airway changes before and after distraction treatment using 3D models. The focus of this study was to examine the change in upper airway volume after distraction osteogenesis and determine which skeletal or soft tissue landmarks can represent airway changes.

According to Ai-Aql et al. (2008) [22], distraction osteogenesis involves three phases: latency, distraction, and consolidation. In the latency period (lasting 3–5 days), a LeFort I osteotomy is performed through the pyriform aperture and pterygomaxillary fissure to downfracture and mobilize the maxilla. As a result, the bilateral Orbitale point might not be changed during osteogenesis and may not affect the orientation of medical CBCT. However, we did not specifically take into account the change in the Orbitale point during osteogenesis when planning the study. The RED frame is then attached to the cranium using bilateral sharp pins. The semirigid surgical wire is attached to the pre-maxilla area and passes through the bilateral subnasale area, but is not yet attached to the horizontal bar. In the distraction period (lasting 10–14 days, depending on the profile), the most common and safe method of producing new immature woven bone is through the application of a tensile force to the callus base at a rate of 1 mm per day. The bone fragments are moved forward by the distraction force. The duration of the treatment depends on the patient’s profile and the skeletal changes. In the consolidation period (lasting 8–12 weeks), after the mobile bone fragments have been moved to the ideal position, it takes time for the osteoid bone to undergo mineralization and final remodeling. During this phase, the distraction device should still be kept passive, and it may be treated with a rigid fixation device to maintain the position. It is important to wait enough time to ensure bone reunion, but potential complications, such as infection, ulcers, and mucosal dehiscence, should be monitored. The external distraction device may cause social and physical hardship, so our patients were surgically treated during summer vacation to reduce their mental stress. The transcutaneous external distraction device is also easier to monitor and handle by the patient’s family than the internal distraction device.

In patients with severe Class III skeletal discrepancies, dentoalveolar compensation and orthognathic surgery are common solutions. However, in patients with cleft lip and palate (CLCP), growth modification methods should also be considered in the treatment plan. By controlling the direction of the RED device, the treatment aims to achieve a clockwise rotation of the mandible, which can improve the concave facial profile. During this stage, the airway volume may be a concern due to the backward rotation of the mandible.

In a study by Chun-Shin Chang et al. in 2017 [15], the changes in airway volume of CLCP patients after undergoing orthognathic surgery were analyzed. The surgery involved maxilla advancement, mandible setback, and clockwise rotation of the maxilla-mandibular complex. The pharyngeal airway was divided into three regions: the velopharyngeal airway (VP), the oropharyngeal airway (OP), and the hypopharyngeal airway (HP). The study found that the volume increased in the VP, decreased in the OP, and remained unchanged in the HP. Although the results showed improvement in snoring, there was no significant effect on sleep-related breathing function. Further research is needed to evaluate the relationship between airway volume changes and the extent of surgical advancement in the maxilla and mandible.

In this study, we discovered that the changes in total airway volume were not all increases. In two cases (Case 7 and 8), the pharyngeal volume changes resulted not only in a decrease in total volume but also in the upper, oropharyngeal, and hypopharyngeal airways. In Case 9, although the total airway volume increased, the oropharyngeal and hypopharyngeal airways showed a decrease. To understand why these cases had different results than others, we measured the changes in FMPA (the angle between the Frankfort horizontal plane and the mandibular plane) and found that the angle on average was −1.08 ± 1.83 degrees (it showed forward rotation in majority), but in Case 7,8,9, the FMPA angle showed an increase (Case 7: +5.0 degrees, Case 8: +14 degrees, Case 9: +8 degrees). The increase in FMPA angle represents a downward and backward rotation of the mandible after treatment, which may have caused the airway volume changes to decrease in these three cases. While it is known from many publications that the position of the mandible effects the size of the airway more than the maxilla, there are articles showing no effect of anterior maxillary protrusion on airway dimension. Therefore, we did not perform a cephalometric analysis of the rotation of the maxillofacial complex (particularly the mandible) in our study. After the TMDO, the maxilla was brought forward by the distraction force, and the forward movement of the mandible was mainly the effect of occlusion adaptation and growth. However, more research is needed to determine the precise relationship between the airway volume changes and the degree of mandibular backward rotation. Another possible factor could be that Cases 7 and 8 were bilateral cleft lip and palate, which may result in more hypertrophic scar formation compared to unilateral CLCP patients, affecting the soft tissue, function, movement, and future facial growth [11,12,13]. As some cases showed extreme values in airway changes (some cases showed negative values in airway volume changes), the mean data of 3D pharyngeal airway volume in our cases showed great deviation (Table 4), especially in the mean value of total airway volume change. The results indicate that there is great variability in airway volume changes in CLCP patients after TMDO treatment, and it is still necessary to evaluate different factors that may influence the airway volume change.

To evaluate the correlation between airway volume changes and the amount of distraction, we compared seven hard tissue landmarks with pharyngeal airway volume changes in 3D images. The results showed a low or even non-correlation between maxillary landmarks (A, ANS, U1) and the changes in the upper, oropharyngeal, and hypopharyngeal airways, and in the total volume change. However, the mandibular landmarks (B, Pog, L1) showed a moderate to high correlation with the changes in the hypopharyngeal airway and total airway volume change (Table 7). The lower correlation between the maxillary landmarks and airway volume changes may be due to the influence of pre-TMDO surgical repair procedures, such as surgical cleft lip repair, gingiva-periosteoplasty, hard palate closure, and alveolar bone grafting on the soft and hard tissue around the Upper and Oropharyngeal airways. The amount of maxillary osteogenesis distraction mainly depends on the patient’s facial profile and maxillary stability.

Limited by the use of 3D graphics in the skull (Figure 4), our study used Point-ANS, A, and U1 to represent the amount of maxillary advancement, but the results showed only negligible to low correlation. The complexity of the maxillary structure in CLCP patients makes it difficult to accurately locate relevant landmarks, and further investigation and software imaging systems that locate different maxillary skeletal landmarks may benefit from the planning and prediction of airway volume changes after maxillary advancement in TMDO treatment and orthognathic surgery.

Cleft deformity not only causes cosmetic problems but also impairs nasal airflow due to distorted anatomy [23,24,25]. Cleft patients have statistically significantly smaller airways compared to non-cleft patients, as shown by Nasal Rhinometry. Although a study by Broadbent et al. [23] found that unilateral cleft children have smaller airways than bilateral cleft children, these differences did not persist over time, and airway volume changes are influenced by multiple factors.

The airway changes were analyzed using CBCT images in this study, which allowed for a more accurate assessment of the changes in airway volume and treatment in three dimensions. The use of CBCT images also enables a more precise analysis of the 3D landmarks, as well as the expansion of soft tissue or craniofacial structures that may be altered by distraction osteogenesis [26,27,28]. Although there may be concerns about radiation exposure and potential image noise due to radiation scattering by metal components of the distraction device, these issues can be reduced by using the smallest field of view that encompasses the region of interest [29,30].

One limitation of the study is the small sample size, as cleft lip and palate patients are rare in Taiwan. Additionally, it can be difficult to control tongue position and breathing phase during CBCT scans [31]. However, we instructed patients to position their tongue in a relaxed state during CBCT scanning. According to a special article by Artese in 2011 [32], the normal resting position of the tongue is one where the tip of the tongue rests on the incisal papilla and the back of the tongue lies along the palate. Therefore, we excluded the effect of tongue position and only discussed changes in the bone points of the mandible. Further studies with larger sample sizes are needed to better understand the differences in function and anatomy between unilateral and bilateral cleft lip and palate patients.

## 6. Conclusions

Our study aimed to evaluate the relationship between airway volume changes and the extent of distraction in Cleft Lip and Cleft Palate (CLCP) patients after undergoing transcutaneous maxillary distraction osteogenesis (TMDO) treatment. The results indicated significant deviation in total airway volume changes and a negligible to low correlation between pharyngeal airway volume changes and 3D distance changes of hard tissue in maxillary landmarks (Point-ANS, A, U1). Conversely, mandibular landmarks (Point B, Pog, L1) showed a higher correlation with airway volume changes, particularly in the hypopharyngeal airway and total airway volume change. Advanced imaging systems may aid in identifying maxillary skeletal landmarks and improve the prediction of airway volume changes after maxillary advancement. However, further investigation is necessary to understand the effect of surrounding soft tissue, such as the tongue, uvula, pharynx size, and upper airway muscle tone, which also play a role in determining pharyngeal airway volume changes in CLCP patients.

## Figures and Tables

**Figure 1 jpm-13-00543-f001:**
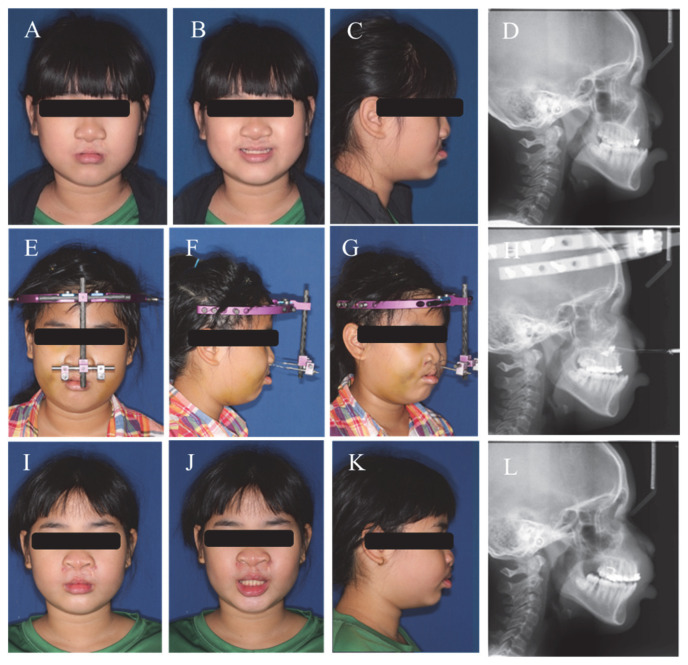
An 11-year-old girl with Left Cleft Lip and Cleft Palate (CLCP) underwent primary lip/palate repair and transcutaneous maxillary distraction osteogenesis (TMDO) to enhance her facial profile and dental alignment (Case-3). The following images provide a visual timeline of her progress: (**A**–**D**) Initial, (**E**–**H**) During TMDO treatment, and (**I**–**L**) two days immediately after TMDO treatment.

**Figure 2 jpm-13-00543-f002:**
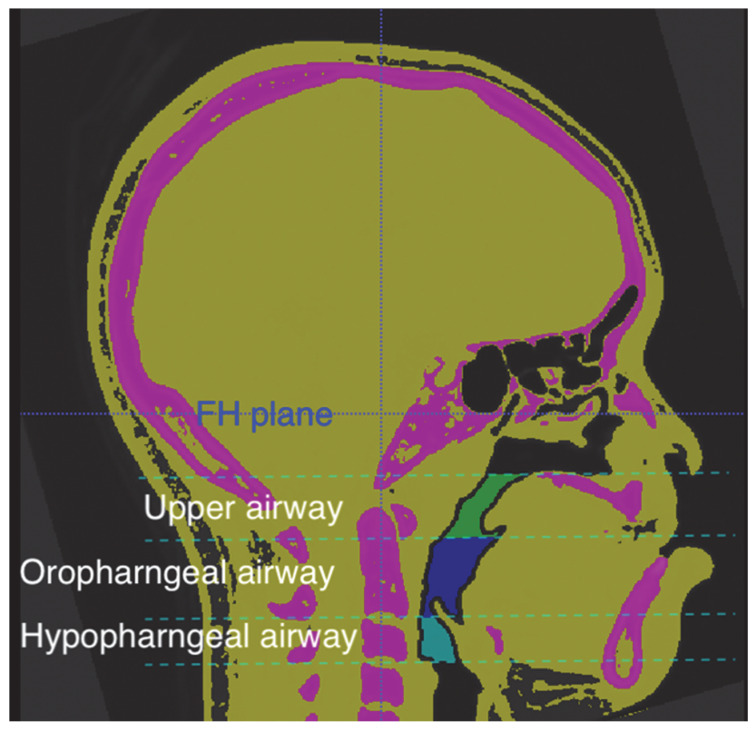
The midsagittal view in Case-3 displays various regions of the segmental airway. The landmarks are defined as follows. (1) Upper airway (in green color): extending from the posterior nasal spine to the lowermost protruding part of C1. (2) Oropharyngeal airway (in navy blue color): extending from the lowermost protruding part of C1 to the lowermost protruding part of C2. (3) Hypopharyngeal airway (in light blue color): extending from the lowermost protruding part of C2 to the lowermost protruding part of C3.

**Figure 3 jpm-13-00543-f003:**
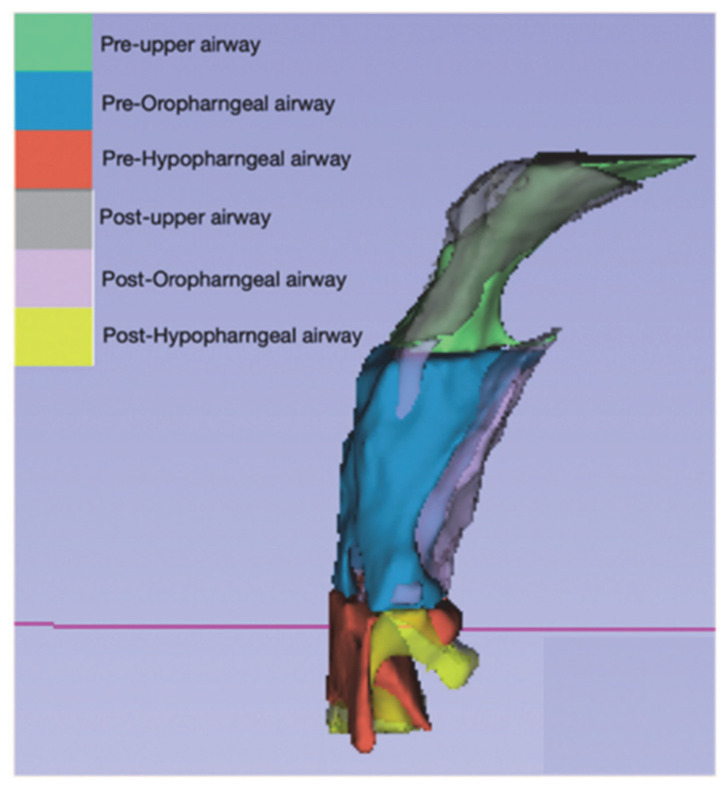
The 3D model images in Figure 2 of Case-3 depict the superimposition of the pharyngeal airway volume. The pharyngeal airway has been divided into three regions namely, the upper airway, the oropharyngeal airway, and the hypopharyngeal airway, each represented by a different color. Our findings indicate that the volume of the pharyngeal airway increased overall after treatment, as seen in the post-treatment images.

**Figure 4 jpm-13-00543-f004:**
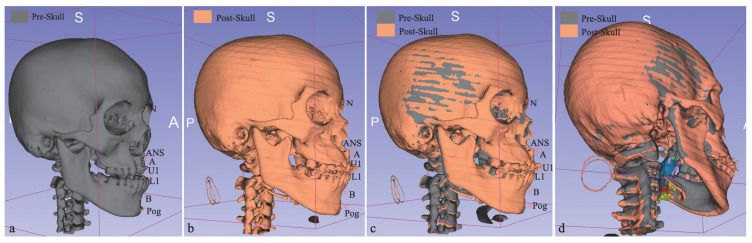
These 3D model images depict the superimposition of pre- and post-skull images, where the pre-skull image shows the 7 points mentioned in (**a**) and the post-skull image shows the 7 points mentioned in (**b**). The overall superimposition image, (**c**), shows the comparison of changes in the skull structure after treatment. (**a**): Pre-Skull, showing 7 points (Point 1: pre-ANS; Point 2: Pre-N; Point 3: Pre-A; Point 4: Pre-B; Point 5: Pre-Pog; Point 6: Pre-U1; Point 7: Pre-L1); (**b**): Post-Skull, showing 7 points (Point 8: Post-ANS; Point 9: Post-A; Point 10: Post-N; Point 11: Post-B; Point 12: Post-Pog; Point 13: Post-U1; Point 14: Post-L1); (**c**): Overall superimposition of skeletal; (**d**): Overall superimposition of airway volume and skeletal displacement.

**Table 1 jpm-13-00543-t001:** Patient Characteristics and Classification.

	Numbers		Total Number of Patients
Boy	7		15
Girl	8		
Bilateral cleft	5		15
Unilateral cleft	right	6	
	left	4	

**Table 2 jpm-13-00543-t002:** Classification and Treatment Duration of the 15 Participants in this Study. Abbreviations: B, Boy; G, girl.

	Gender	Age	Cleft Type	Period of Distraction (Days)	Period of Maintain (Days)
Case-1	G	10.9	right	22	65
Case-2	G	10.9	right	15	72
Case-3	B	11.3	left	29	63
Case-4	B	11.8	left	45	57
Case-5	B	11.5	bilateral	28	89
Case-6	G	11.5	left	34	86
Case-7	B	12.4	bilateral	22	84
Case-8	B	11.9	bilateral	36	51
Case-9	G	11.4	right	23	77
Case-10	G	11.3	right	24	67
Case-11	G	11.7	right	48	72
Case-12	G	11.3	right	26	66
Case-13	G	11.2	bilateral	36	32
Case-14	B	10.8	left	55	86
Case-15	B	11.6	bilateral	15	63
Mean		11.43 ± 0.42		30.53 ± 11.78	68.67 ± 15.17

**Table 3 jpm-13-00543-t003:** Definitions of Landmarks (Houston, 1983) [21].

Skeletal Landmarks	
N (nasion)	The most anterior point on the frontonasal suture
ANS (anterior nasal spine)	The tip of the anterior nasal spine
Point A (subspinale)	The deepest midline point in the curved bony outline from the base to the alveolar process of the maxilla
Point B (supramentale)	The most posterior point on the outer contour of the mandibular process in the median plane
Pog (pogonion)	The most anterior point of the bony chin in the median plane
**Dental landmarks**	
U1 (incisor superius)	The tip of the crown of the most anterior maxillary central incisor
L1 (incisor inferius)	The tip of the crown of the most anterior mandibular central incisor

**Table 4 jpm-13-00543-t004:** Changes in Pharyngeal Airway Volume (mm^3^) as analyzed in 3D Models.

	T0	T1	T1-T0	
	Mean ± SD	Mean ± SD	Mean ± SD	*p* Value
A	2736.07 ± 1103.49	3682.40 ± 1914.45	946.33 ± 1628.85	0.041 *
B	2269.41 ± 868.90	3360.73 ± 1331.10	1091.32 ± 1032.81	0.001 **
C	945.71 ± 335.18	1443.11 ± 582.47	497.41 ± 675.24	0.013 *
Total volume change	1983.73 ± 1.118.72	2828.75 ± 1684.44	845.02 ± 1180.70	

Abbreviation: T0, pre-treatment (1.5 months before starting distraction pre-treatment); T1, post-treatment (6–8 months after removal distraction appliances); A, Upper airway(mm^3^); B, Oropharyngeal airway(mm^3^); C, Hypopharyngeal airway(mm^3^); detailed interpretation of the airway can be seen in Figure 2. * *p* value <0.05; ** *p* value <0.005.

**Table 5 jpm-13-00543-t005:** Individual Case Analysis of Changes in Pharyngeal Airway Volume (mm^3^) using 3D Models.

	A			B			C			
	T0	T1	T1-T0	T0	T1	T1-T0	T0	T1	T1-T0	Total Volume Changes
Case-1	3475	5915	2440	1990	3319	1329	1014	1068	54	3823
Case-2	1395	1898	503	1166	2110	944	767.4	2015	1247.6	2694.6
Case-3	2116	2229	113	3488	3700	212	1222	1558	336	661
Case-4	3831	2279	−1552	1676	4057	2381	791.7	1377	585.3	1414.3
Case-5	3723	6671	2948	3240	5198	1958	572.7	1443	870.3	5776.3
Case-6	4258	7070	2812	3443	5955	2512	1343	1948	605	5929
Case-7	2891	2169	−722	2431	2164	−267	1315	1174	−141	−1130
Case-8	3578	2678	−900	2250	1487	−763	950.1	874.6	−75.5	−1738.5
Case-9	3717	5093	1376	3253	3056	−197	896.1	812.2	−83.9	1095.1
Case-10	2858	2871	13	2595	3679	1084	1370	1771	401	1498
Case-11	603	1391	788	2545	3659	1114	1428	991.1	−436.9	1465.1
Case-12	1846	5128	3282	2351	4965	2614	580.7	2891	2310.3	8206.3
Case-13	1493	2016	523	835.6	1523	687.4	449.4	783.8	334.4	1544.8
Case-14	1851	5028	3177	1794	3343	1549	993.7	1901	907.3	5633.3
Case-15	3406	2800	−606	983.6	2196	1212.4	491.8	1039	547.2	1153.6
Mean(mm^3^)	2736.07 ± 1103.49	3682.40 ± 1914.45	946.33 ± 1628.85	2269.41 ± 868.90	3360.73 ± 1331.10	1091.32 ± 1032.81	945.71 ± 335.18	1443.11 ± 582.47	497.41 ± 675.24	2535.06 ± 2791.80

Abbreviation: T0, pre-treatment (1.5 months before starting distraction pre-treatment); T1, post-treatment (6–8 months after removal distraction appliances); A, Upper airway(mm^3^); B, Oropharyngeal airway(mm^3^); C, Hypopharyngeal airway(mm^3^); detailed interpretation of the airway can be seen in Figure 2.

**Table 6 jpm-13-00543-t006:** Changes in the 3D Distance of Seven Landmarks between Pre-Treatment and Post-Treatment.

	T1–T0
	Mean ± SD
ΔN	0.82 ± 0.23
ΔANS	9.85 ± 3.60
ΔA	14.22 ± 4.57
ΔU1	16.77 ± 4.60
ΔB	3.33 ± 1.60
ΔPog	3.54 ± 1.84
ΔL1	4.83 ± 1.37

Abbreviation: T0, pre-treatment (1.5 months before starting distraction pre-treatment); T1, post-treatment (6–8 months after removal distraction appliances); Δ, represents the distance change from T1 to T0 (all value present positive value); N, nasion; ANS, anterior nasal spine; Point A, subspinale; Point B, supramentale; Pog, pogonion, U1, incisor superius, L1, incisor inferius (detailed definition can be seen in Table 3.

**Table 7 jpm-13-00543-t007:** Correlation between Changes in 3D Pharyngeal Airway Volume (mm^3^) and Changes in 3D Distance of Seven Landmarks.

	Upper Airway	Oropharyngeal Airway	Hypopharyngeal Airway	Total Volume Change
N	−0.11 *	−0.02 *	0.00 *	0.00 *
ANS	0.18 *	0.07 *	−0.01 *	−0.01 *
A	0.04 *	0.12 *	−0.13 *	−0.12 *
U1	−0.03 *	0.04 *	−0.18 *	−0.18 *
B	0.18 *	0.37 **	0.76 ****	0.77 ****
Pog	0.43 **	0.45 **	0.75 ****	0.75 ****
L1	0.08 *	0.45 **	0.53 ***	0.53 ***

The detailed 3D distance changes of seven landmarks can see in Table 6 and three regions airway volume changes can be seen in Table 4 Based on these data, we derived the following correlating results. Abbreviation: N, nasion; ANS, anterior nasal spine; Point A, subspinale; Point B, supramentale; Pog, pogonion, U1, incisor superius, L1, incisor inferius. Interpretation of Pearson correlation: * Negligible correlation (0.00~0.30); ** Low positive correlation (0.30~0.50); *** Moderate positive correlation (0.50~0.70); **** High positive correlation (0.70~0.90).

## Data Availability

The datasets supporting the conclusions of this article are included within the article.

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
