# Peer review of "Assessing the Impact of Transcutaneous Maxillary Distraction Osteogenesis on Pharyngeal Airway Volume in Children with Cleft Lip and Palate: A 3D Evaluation Study"

_jpm, 2023, doi:10.3390/jpm13030543_

Round 1

Reviewer 1 Report

- Methodology should be explained in more detail. 

- Inclusive and exclusive criteria should be mentioned.

- Probands with CLCP were syndromic or non-syndromic? 

Reviewer 2 Report

The work given to me for evaluation concerns the assessment of the impact of distraction osteogenesis on the size of the airways. Only 15 cases are presented in the paper, which is not a sufficiently large group in statistical terms. In addition, I noted:

Material and methods

Table 2 is repeated

There is no information in which standing/lying position CBCT of the patient's head was performed?

Change of the Or point during osteogenesis - this is how CBCT was oriented, whether it was taken into account when planning the study

It is known from many publications that the position of the mandible affects the size of the airway more than the jaw. There are articles showing no effect of anterior maxillary protrusion on airway dimensions. Has a cephalometric analysis of the rotation of the maxillofacial complex, in particular the mandible, been performed?

Discussion:

 The first paragraph should be in the introduction, the second paragraph in materials and methods.

The paper does not mention that the position of the tongue has a great influence on the size of the airways, which directly depends on the position of the mandible - therefore, a positive correlation with bone points in the mandible, despite the fact that the distraction concerned the jaw

In the discussion, the authors mention the measurement of the FMPA angle when discussing three cases describing the reduction of airway dimensions. The result of this measurement is not given in results. Analysis of the airway width with the position and rotation of the mandible could yield more interesting results and relationships. I suggest adding these items in the Materials and Methods section.

There is no paragraph comparing other works in which the authors examine airway dimension v jaw displacement. I propose to add, as it is the most important from the point of view of this work.

Round 2

Reviewer 2 Report

Thank you for taking my comments into consideration